# E-Cigarette Aerosol Condensate Leads to Impaired Coronary Endothelial Cell Health and Restricted Angiogenesis

**DOI:** 10.3390/ijms24076378

**Published:** 2023-03-28

**Authors:** Michael Chhor, Esra Tulpar, Tara Nguyen, Charles G. Cranfield, Catherine A. Gorrie, Yik Lung Chan, Hui Chen, Brian G. Oliver, Lana McClements, Kristine C. McGrath

**Affiliations:** 1School of Life Sciences, Faculty of Science, University of Technology Sydney, Sydney, NSW 2007, Australia; michael.chhor@student.uts.edu.au (M.C.); esra.s.tulpar@student.uts.edu.au (E.T.); tara.nguyen@uts.edu.au (T.N.); charles.cranfield@uts.edu.au (C.G.C.); catherine.gorrie@uts.edu.au (C.A.G.); yik.chan@uts.edu.au (Y.L.C.); hui.chen-1@uts.edu.au (H.C.); brian.oliver@uts.edu.au (B.G.O.); 2Institute for Biomedical Materials and Devices, Faculty of Science, University of Technology Sydney, Sydney, NSW 2007, Australia

**Keywords:** e-vaping, cardiovascular disease, smoking, nicotine, atherosclerosis

## Abstract

Cardiovascular disease (CVD) is a leading cause of mortality worldwide, with cigarette smoking being a major preventable risk factor. Smoking cessation can be difficult due to the addictive nature of nicotine and the withdrawal symptoms following cessation. Electronic cigarettes (e-Cigs) have emerged as an alternative smoking cessation device, which has been increasingly used by non-smokers; however, the cardiovascular effects surrounding the use of e-Cigs remains unclear. This study aimed to investigate the effects of e-Cig aerosol condensate (EAC) (0 mg and 18 mg nicotine) in vitro on human coronary artery endothelial cells (HCAEC) and in vivo on the cardiovascular system using a mouse model of ‘e-vaping’. In vitro results show a decrease in cell viability of HCAEC when exposed to EAC either directly or after exposure to conditioned lung cell media (*p* < 0.05 vs. control). Reactive oxygen species were increased in HCAEC when exposed to EAC directly or after exposure to conditioned lung cell media (*p* < 0.0001 vs. control). ICAM-1 protein expression levels were increased after exposure to conditioned lung cell media (18 mg vs. control, *p* < 0.01). Ex vivo results show an increase in the mRNA levels of anti-angiogenic marker, *FKBPL* (*p* < 0.05 vs. sham), and endothelial cell adhesion molecule involved in barrier function, *ICAM-1* (*p* < 0.05 vs. sham) in murine hearts following exposure to electronic cigarette aerosol treatment containing a higher amount of nicotine. Immunohistochemistry also revealed an upregulation of FKBPL and ICAM-1 protein expression levels. This study showed that despite e-Cigs being widely used for tobacco smoking cessation, these can negatively impact endothelial cell health with a potential to lead to the development of cardiovascular disease.

## 1. Introduction

Cardiovascular diseases (CVD) and the resultant vascular complications are a major cause of mortality, accounting for 31% of all deaths worldwide [1,2]. The development of CVD is multifactorial and has been associated with risk factors including tobacco cigarette smoking, obesity, high cholesterol, and high blood pressure [2,3]. Notably, 10% of all CVD cases are attributable to smoking tobacco cigarettes [4]. Depending on an individual’s frequency and habit, smoking can increase the risk by at least two-fold for developing conditions including heart failure and acute myocardial infarction (AMI) compared to the other risk factors [5]. Additionally, it is reported that smoking can act synergistically with other risk factors such as hypertension and diabetes mellitus in multiplying the level of risk for CVD development [6].

Electronic cigarettes (E-Cigs) have recently emerged as a supposedly less toxic and less carcinogenic alternative to traditional cigarettes without any combustion [7]. E-Cigs are electronic devices that can differ in design between brands; however, they are generally composed of a rechargeable battery, an e-liquid tank (with thousands of potential flavouring) and an atomiser element that heats and aerosolises the e-liquid to create a vapour for smoking. The e-liquid is comprised of propylene glycol (PG), vegetable glycerin (VG), and, optionally, nicotine. There is also a large market for different flavouring [8,9,10]. E-Cigs use has been traditionally perceived as harmless, with recent trends showing an increase in usage amongst current smokers, but additionally, non-smokers and young adolescents [7,11]. Studies have reported the presence of carbonyl compounds in e-Cig aerosols, notably: formaldehyde, acetaldehyde, and acrolein, as well as long-chain and cyclic alkanes and alkenes [12]. Additionally, trace amounts of metals have been reported, such as aluminum, barium, chromium, and cadmium within the e-Cig aerosol [10,13]. These chemicals are known to be harmful and cytotoxic, causing pulmonary and cardiovascular stress [14]. Whilst these chemicals have been reported to be lower in concentration from their traditional tobacco cigarette counterparts, there remain many other residual chemicals generated during the heating process in addition to the role of nicotine that could contribute to early atherogenesis [15].

Endothelial cells play an important role in cardiovascular homeostasis, regulating the permeability of the arterial vessels, and are the first responders to inflammatory stimuli [16]. Endothelial dysfunction (ED) is an early critical event that leads to atherosclerosis and heart failure, affecting vascular integrity through reduced vasodilation, increased inflammation, and prothrombic activity [17,18]. Experimental studies have demonstrated that exposure to the harmful chemicals generated from tobacco smoke not only results in vascular dysfunction, but also leads to the activation of the vascular endothelium as a result of a shift to a pro-oxidative state and increased expression of adhesion molecules on the surface of endothelial cells—an early event in atherosclerosis [19,20].

FK506 binding protein-like (FKBPL), an anti-angiogenic protein and key determinant of CVD, was shown to be increased in human plasma as a result of smoking [21]. FKBPL is secreted by endothelium, and when knocked down in mice, it leads to endothelial dysfunction and impaired vascular integrity [22], suggesting that angiogenic balance is the key to maintaining healthy endothelium. CD31/PECAM1 is an endothelial cell adhesion and signalling molecule that mediates both homophilic and heterophilic adhesion in angiogenesis [23,24]. Increased levels of CD31 have also previously been associated with early COPD and cardiovascular complications as a result of smoking [25,26].

While e-Cigs have been considered a safe alternative to conventional cigarettes, their potential as a smoking cessation device remains controversial. Moreover, of concern is the rising usage of e-Cigs by adolescents and young adults who were never exposed to tobacco cigarettes. This is concerning given that the safety profile of e-Cigs is still unknown, including its impact on the cardiovascular system. Therefore, in this study, we aimed to determine the impact of e-Cigs aerosol condensate (EAC) on endothelial cell homeostasis through the assessment of its effects on the viability of human coronary artery endothelial cells (HCAECs). We further investigate EAC’s contributions to endothelium inflammation, oxidative stress, and angiogenesis as part of the mechanisms implicated in this effect. Finally, the immediate impact of nicotine on cell membrane ion permeability was demonstrated using a tethered bilayer lipid membrane (tBLM) assay. The expression of key inflammatory endothelial cell (ICAM-1 and VCAM-1) and angiogenesis markers (FKBPL and CD31) were also assessed ex vivo in hearts from mice exposed to e-Cigs aerosol in vivo. It is hypothesized that EAC and e-cigarette aerosols will affect endothelial cell health, increasing the expression of inflammatory and anti-angiogenic markers related to endothelial dysfunction and the pathogenesis of cardiovascular disease.

## 2. Results

### 2.1. Exposure of HCAEC to EAC-Treated Lung Cell Conditioned Media Results in Cytotoxicity

HCAEC directly exposed to 4% and 8% EAC generated from PG/VG without flavouring or nicotine showed a significant reduction in cell viability to 34 ± 8.9% (*p* < 0.001)/47 ± 10.9% (*p* < 0.05) and 29 ± 4.9% (*p* < 0.01)/38 ± 2.2% (*p* < 0.01) compared to the control cells, respectively (Figure 1A). A decrease in cell viability to 46 ± 6.2% (*p* < 0.05 versus control cells) was shown for HCAEC exposed to tobacco flavour EAC generated from e-liquid without (0 mg/mL) nicotine at the more concentrated EAC of 8%.

Using e-Cigs, the aerosol first comes into contact with the lung epithelial cells before influencing endothelial cells. Thus, to determine whether the response to the EAC from lung epithelial cells would affect the viability of HCAEC, A549 epithelial lung cells were exposed to EAC for 24 h before the conditioned media was used to treat HCAEC for another 24 h. Similar to the response of HCAEC directly exposure of EAC, exposure of conditioned lung epithelial cell media exposed to 4 and 8% EAC generated from PG/VG without flavouring or nicotine resulted in significantly reduced HCAEC viability (Figure 1B). Exposure of conditioned lung epithelial cell media exposed to EAC generated from tobacco flavoured e-liquid without nicotine (0 mg/mL) also resulted in a decrease in cell viability to 32 ± 4.9% (*p* < 0.001) compared to the control cells (Figure 1B). 

Whilst the MTT assay is a widely used assay for detecting cellular toxicity, there are confounding variables that should be considered when performing the assay [27]. To assess if our EAC could reduce MTT, we performed an MTT assay to determine if there were any interference of the MTT dye with the EAC. The results show no difference in absorbance for the lower concentrations of EAC, except PG/VG at 2% where a significant increase in absorbance from 1.0 (ctrl) to 1.09 (** *p* < 0.01; Appendix A) was observed. A significant increase was also observed for 8% EAC 0 mg and 18 mg with absorbance of 1.2 and 1.1, respectively (** *p* < 0.0001 vs. Ctrl). This suggests a minor catalytic effect of EAC on MTT reduction that is mediated by EAC. 

### 2.2. Direct Exposure to EAC or Indirectly to EAC-Lung Cell Conditioned Media Induces ROS Levels

ROS have been shown to play a crucial role in inducing endothelial dysfunction and oxidative stress in cells, a key mechanism behind atherogenesis and heart failure [28,29]. HCAEC exposed directly to 8% EAC generated from PG/VG or tobacco flavour e-liquid with (18 mg/mL) nicotine solution showed an increase in ROS levels by ~7.5-fold (*p* < 0.01) compared to control (Figure 2A).

Given the results of the cell viability experiments (Figure 1A), we had selected 2% EAC, as this did not result in a significant reduction of cell viability following direct exposure for PG/VG and 4% EAC to assess for effects on the ROS levels produced by HCAEC in co-culture conditions. Similar to the results observed in monoculture, an increase in ROS levels were shown in the co-culture model for HCAEC exposed to lung epithelial cell conditioned media for 4% PG/VG EAC, 4% tobacco flavoured EAC with (18 mg), or without nicotine (0 mg) by 6.7-fold, 3.2-fold, and 3.5-fold compared to the control, respectively (*p* < 0.0001; Figure 2B). HCAEC exposed to lung cell conditioned media showed a significant increase in ROS levels for 2% PG/VG EAC and 2% tobacco flavoured EAC without nicotine (0 mg) by 2.7-fold and 2.6-fold compared to the control, respectively (*p* < 0.0001; Figure 2B). No significance was shown for 2% tobacco flavoured EAC with (18 mg).

### 2.3. Adhesion Molecule Expression Increases in HCAEC after EAC Exposure for ICAM-1, but Not VCAM-1

A critical early event in atherogenesis is the adhesion of monocytes to the endothelium. The adhesion of monocytes occurs when the endothelial cells become activated in response to several factors, including oxidative stress, which leads to the upregulation of cell adhesion molecules (CAMs), such as VCAM-1 and ICAM-1 [17]. VCAM-1 or ICAM-1 protein levels were not significantly changed in HCAEC monoculture regardless of EAC used (Figure 3A,B). Although no significance was shown, an increase in ICAM-1 protein expression level to 60 ± 21.1% (*p* = 0.068 compared to control) could also be observed for HCAEC directly exposed to 2% EAC generated from e-liquid containing 18 mg/mL (Figure 3B). Given the monoculture showed a strong trend to changes for ICAM-1 protein levels with nicotine at 2% EAC, next, we only assessed the ICAM-1 levels in the co-culture model using 2% EAC. In contrast to the results of HCAEC directly exposed to EAC, exposure of HCAECs to conditioned media from lung epithelial cells treated with 2% EAC generated from e-liquid containing 18 mg/mL, an 83 ± 8.9% (*p* < 0.01 compared to control) increase in ICAM-1 protein levels was observed (Figure 3C). 

### 2.4. EAC from Nicotine Containing e-Liquid Alters Membrane Permeability

Given nicotine is known to be membrane-permeable, we next assessed if EAC has an effect on membrane permeability using a tethered bilayer lipid membranes (tBLMs) assay [30]. These tBLMs are a model cell membrane anchored to a gold electrode that, when used in conjunction with electrical impedance spectroscopy techniques, enable a measure of how compounds and solutions can alter membrane structure and permeability to ions [31]. We tested 1% and 10% EAC generated from e-liquid with and without nicotine on tBLMs and measured the effects on membrane ion permeabilizaton using electrical impedance spectroscopy (Figure 4A). When the EAC is sourced from a fluid containing 18 mg/mL nicotine and applied to the tBLM, there is a marked increase in membrane conduction as measured using electrical impedance spectrocopy.

In contrast to the change in membrane conduction, the membrane capacitance does not show similar changes (Figure 4B). Membrane capacitance is a measure of membrane thickness and/or water content [31]. These data suggest that the EACs are not causing any significant membrane structural changes. 

### 2.5. E-Cigarette Aerosol Increases ICAM-1 mRNA Expression in Murine Hearts

Adhesion molecules play a critical role in the pathogenesis of atherosclerosis, embedded with the inflammatory and immune response [32]. Systemic inflammation is a pivotal process of atherosclerosis and similarly contributes to the implication of endothelial cell activation in the pathogenesis of developing heart failure [33]. We therefore assessed the expression of adhesion molecules in animals exposed to e-Cig aerosol with or without nicotine. A significant difference in the mRNA expression of *ICAM-1* and *FKBPL* levels were shown between the SHAM and 18 mg nicotine groups and SHAM and 0 mg nicotine groups, respectively (Figure 5B,C, *p* < 0.05). Contrastingly, the mRNA expression of *VCAM-1* and *CD31* exhibited no significant difference between groups (Figure 5A,D). 

### 2.6. Cardiac Angiogenesis Markers Are Dysregulated by E-Cig Aerosol Exposure

Angiogenic impaired regulation is an integral process in the development of cardiovascular diseases and therapeutic interventions. We therefore assessed FKBPL and CD31 protein expression in the LV of mice exposed to e-Cig aerosol with or without nicotine. Whilst no significant change in *FKBPL* or *CD31* mRNA expression was observed, immunohistochemistry showed a significant 10-fold increase in FKBPL protein in 18 mg nicotine treatment group (*p* < 0.01) (Figure 6B) compared to the SHAM group. CD31 level paralleled the trend of FKBPL protein expression, where a significant 1.7-fold increase was seen in the 18 mg nicotine treatment group (*p* < 0.05) (Figure 6C). 

## 3. Discussion

The goal of the present study was to assess the effects of the use of e-Cigs on the health of endothelial cells. Our in vitro studies show endothelial cells exposed directly to EAC generated from the base e-liquid solution (PG/VG), e-liquid solution with or without nicotine induced a decrease cell viability, and an increase in ROS levels. Importantly, our study is the first to show that these adverse effects were exacerbated or remained even after exposure to lung cells using our indirect co-culture-like treatment model. In vivo, cardiac changes indicative of angiogenesis was observed in animals, albeit only in animals exposed to e-Cig aerosol containing nicotine. These findings suggest e-Cigs can modulate and induce adverse changes to endothelial cells and the heart.

We wanted to evaluate the effects of e-Cig vaping where the e-liquid is heated through a device to generate aerosol that is subsequently inhaled by the user. Current studies vary in the methodologies used to collect and use e-Cig aerosol [14]. In this study, we chose to collect the condensate from the e-Cig aerosol to evaluate their effects on the health of endothelial cells at varying concentrations. Many studies exhibit the effect of e-Cig aerosol in individual cultures of a single cell type in which they can examine, for example, the respiratory tract or the endothelial effect [34]. In this study, we used both A549 epithelial lung cells and HCAECs to emulate the process of contacting the epithelial layer of the lung first before the e-Cigs metabolites reach the endothelial cells in the blood vessel. The exposure conditions used are based on previous studies within the same institute (UTS [35]). Tobacco flavouring was chosen due to its popularity amongst cigarette smokers [35] and relatively low ROS [36] content compared to its flavoured alternatives, and it is also the only flavour approved by the FDA [37]. Commercially available e-liquids can range from nicotine concentration of 0 mg/mL up to a concentration of 24 mg/mL, where 10 mg/mL appears to be the median amount for most users [38,39]. The chosen nicotine dose of 18 mg/mL is reflective of light smokers based on previously measured plasma cotinine levels [40,41]. Together, these treatment groups provide a reflective model of human e-Cig use, and importantly, our study shows that endothelial cells and markers of cardiac health are affected by e-Cig aerosol both in vitro and in vivo.

In this study, we demonstrated a significant reduction in cell viability of HCAECs following direct exposure to EAC generated from the e-liquid base constituents, PG and VG, alone. Noticeably, cell viability is shown to be decreased in all treatment groups, regardless of nicotine or flavouring, particularly using 8% EAC. This is consistent with the observations in the in vivo studies, where the effects on the lung, kidney, and liver seem to be nicotine-independent, suggesting the toxicity of heated base constituents and other mechanical factors, such as device settings, in the aerosolisation product [40,41,42]. The cytotoxic effect of PG/VG may be attributed to the thermal decomposition of the components, which produce toxic carbonyl compounds that are similarly present in cigarette smoke [43,44,45]. It was found that even the PG/VG treatment, absent of both flavouring and nicotine, is cytotoxic towards endothelial cells and possibly more so than the other treatment groups. Our results are in alignment with Anderson et al. (2016) [7] and Putzhammer et al. (2016) [46], who similarly showed significantly reduced cell viability in human umbilical vein endothelial cells (HUVEC) exposed to tobacco flavour and a variety of e-liquids. Of interest in our study, however, is that we showed significant cytotoxic effects in HCAECs exposed to conditioned media from lung cells exposed to EAC generated from the base/tobacco e-liquid (with or without nicotine), indicating the EAC likely initiate pro-inflammatory conditions in lung epithelial cells that subsequently induced a detrimental effect on the HCAECs.

Oxidative damage as a result of an imbalance in antioxidants and ROS levels has been shown to play an important role in atherogenesis and endothelial dysfunction during cigarette smoking [47,48,49]. In this study, we showed that HCAEC exposed directly to EAC at high concentrations with or without nicotine resulted in increased ROS levels in endothelial cells compared to the controls. Our results corroborate with previous studies, which showed e-Cig vapour extracts increased levels of ROS expression in varying types of endothelial cells and that the pre-treatment of antioxidants on cells abrogated this effect [7,46,50,51]. Nitric oxide (NO) generated by endothelial nitric oxide synthase (eNOS) plays a crucial role in maintaining vascular physiology. In an oxidative stress state, eNOS uncoupling occurs, which results in ROS rather than NO being produced, cascading into the production of peroxynitrite (ONOO^-^) that has oxidative and cytotoxic effects, exacerbating endothelial dysfunction [52]. Whilst we did not assess if EAC induced eNOS uncoupling in HCAECs, El-Mahdy et al. recently demonstrated in situ induction of Nox-dependent ROS production and uncoupling of endothelial NO synthase by e-Cig exposure [53]. Decreased cell viability was similarly observed with HCAECs exposed to conditioned media from lung cells, even at small doses of 2% EAC, demonstrating significantly increased ROS levels. Whether lung cells exposed to EAC result in an increase in the secretion of pro-inflammatory cytokines and therefore induce further adverse effects on the HCAECs requires further investigation. 

The oxidative stress response is linked to the inflammatory pathway, both of which lead to a disruption in the endothelial equilibrium and subsequently endothelial dysfunction, pivotal in the early stages of atherosclerosis. The first step in endothelial dysfunction is the expression of molecules that aid in the adhesion of monocytes to the endothelium and subsequent migration into the subendothelial space [54]. Whilst no change in VCAM-1/ICAM-1 protein expression was observed following direct EAC treatment, indirect EAC treatment induced an increase in ICAM-1 in the HCAEC. In line with results from our study, a study by Makwana et al. (2021) [55] showed a significant increase in ICAM-1 expression in human aortic endothelial cells (HAECs) within a cardiovascular microfluidic model was reported following treatment with traditional cigarette conditioned media, but not e-Cig conditioned media. Makwana et al., (2021) [55] also determined a significant e-Cig aerosol-induced (at the highest dose) increase in THP-1 monocyte adhesion to HAECs albeit only within 10 min of the adhesion period that diminished over time; the effect of tradition cigarette condition media was more pronounced at longer time points. Similarly, Muthumalage et al. (2017) [56] found significant dose-dependent increases in the pro-inflammatory cytokine, IL-8, following in vitro treatment of monocytic cells with flavoured e-liquid. IL-8 and ICAM-1 are, respectively, chemoattractant and adhesion molecules that are involved in monocyte adhesion [52]. However, it is noted that expression of these molecules can be dependent on the specific cell and stimuli type. It is noted that ROS generation reportedly increases *ICAM-1* transcription in endothelial cells, but not always in epithelial cells [57]. This presents a possible ICAM-1 specific role in adhesion regulation after exposure to e-Cig condensate in endothelial cells. However, further investigation is required such as a monocyte adhesion assay that was performed by Makwana et al. (2021) [55] to determine the direct and indirect effect of EAC on THP-1 adhesion to HCAEC. Nevertheless, the assessment of murine hearts obtained from an in vivo model where mice were exposed to e-Cig aerosol with or without nicotine for 12 weeks showed an increase in cardiac ICAM-1 protein levels in mice exposed to e-Cig aerosol containing nicotine, suggesting that in vivo ICAM-1 could be initiating these early atherosclerosis changes. Nicotine has been demonstrated to have anti-inflammatory properties, suggesting that other factors, such as flavouring or the combination of both, are responsible for the increased inflammatory response [58].

In relation to angiogenesis, although changes were observed at the mRNA level only with 0 mg nicotine, FKBPL at the protein level was significantly increased following exposure to nicotine e-Cig aerosol (18 mg). Similarly, CD31 [21,59] was also increased following exposure to e-Cig aerosol with nicotine, perhaps as part of the compensatory mechanism. The changes at the mRNA and protein levels are not always aligned, and it is well-known that FKBPL undergoes post translational modification due to its co-chaperone role [22,23]. Both FKBPL and CD31 related phenotypical changes are due to the changes at the protein level rather than the mRNA level. Hence, these results are more relevant to the downstream effects than the mRNA levels. The determinant factor for these results appears to involve the presence of nicotine, which has been shown to have pro-angiogenic properties [13]. Nicotine exhibits dose-dependent impacts on endothelial cell homeostasis and exhibits angiogenic effects that may be responsible for the pathogenesis of diseases like atherosclerosis [5,15,43]. Nicotinic acetylcholine receptors (nAChRs) are ligand-gated cation channels abundant in endothelial cells and mediate functions, such as proliferation, migration, and angiogenesis in vivo [60]. The effects of nicotine binding to these receptors include endothelium vasodilation, reduced NO availability and eNOS uncoupling, and directly acting on the elements involved in plaque formation [52,61]. Increases in FKBPL as a key anti-angiogenic regulator [62,63] and CD31 in the presence of nicotine are indicative of restrictive angiogenesis and perhaps a compensatory increase in the number of endothelial cells [22], suggesting that the combination of e-Cigs with nicotine are damaging to the cardiac vasculature causing early endothelial cell damage. This was also demonstrated in vitro. Furthermore, using our tethered membrane conductance platform, it was determined that nicotine is capable of altering the permeability of lipid bilayers to ions, such as Na^+^, which would have implications for a cell’s ability to maintain membrane potential homeostasis. The nicotine was also readily washed from the membrane, suggesting it has a rapid off-rate, as predicted by its membrane–water partition coefficient [64]. This is consistent with the rapid “hit” that smokers might feel upon initial nicotine exposure, which then rapidly falls away. Ultimately, nicotine plays a critical role in cell migration and vascular permeability, all of which can stimulate the development of atherosclerotic CVD [61].

Whilst we did not determine the exact mechanistic pathway of e-Cig aerosol that led to the adverse effects on endothelial health, we show that tobacco flavouring and nicotine can affect the extent of these adverse effects. However, this is the first study that implicates a critical anti-angiogenic protein, FKBPL, in the EAC-induced endothelial/heart damage. Unlike our in vivo results, our in vitro findings suggest that e-Cig aerosols affect endothelial homeostasis independent of nicotine. It has been shown that endothelial cell sensitivity of particulates, independent of nicotine, can elicit pro-inflammatory responses that disrupt endothelial cell homeostasis and progress CVD pathogenesis [16].

To the best of our knowledge, this is the first study to demonstrate the disruption of endothelial homeostasis following exposure to conditioned media from lung epithelial cells exposed to EAC, which seems to be more pronounced than direct EAC-exposure. This result is significant as it demonstrates that e-Cig use can potentially lead to the activation of endothelial cells, even after the EAC undergoes first-pass metabolism by lung epithelial cells. We are also reporting, for the first time, changes in a key anti-angiogenic mechanism mediated through FKBPL in murine hearts following exposure to E-cig aerosol with nicotine, suggesting that this combination can lead to cardiac damage and diastolic dysfunction, which we have previously shown in human studies where FKBPL was increased in the presence of diastolic dysfunction [21].

The limitations of this research article may be attributed to the wide and unregulated nature of the e-Cig market. We only used one flavour, one dose, and one e-Cig device in the animal modelling. There are thousands of e-Cig liquid flavours available in the market, and the by-products and constituents of the e-liquid differ between flavours. We only chose a relatively low dose exposure seen in light smokers, which cannot represent the situation of heavy smokers. Similarly, the e-Cig device market has grown exponentially in recent years, where different generations and styles of devices will contain varying atomiser strengths that can affect the aerosolisation process and chemical products of the e-liquids. We also used a different source of PG/VG mixture from that used in the commercial e-liquid and did not determine the exact amount of nicotine in our in vitro experiments. We used the conditioned media from A549 cells following exposure to EAC to assess if there are any metabolites from the A549 that could subsequently affect the HCAEC, the results may be an effect of unsuitable culture medium. Nevertheless, the control cells in the indirect co-culture like model were exposed similarly to conditioned media therefore any further effect from the EAC could still be observed. Future studies should examine the use of transwell membrane to confirm the results. We also note that the absorbance reading with incubation of the EAC with MTT reagent alone indicated interference of the EAC with the MTT assay. Therefore, future studies should ensure EAC-only controls are included when performing the MTT assay in addition to using complementary assays, such as the lactate dehydrogenase assay or live/dead staining to confirm the results. A further limitation of the study is that it remains largely descriptive of the effects of e-Cig extract, and more in-depth future studies addressing FKBPL-related molecular mechanism should be performed. We believe that these limitations must be taken into consideration in future studies. Additionally, there is no direct comparison of EAC to the effects of tobacco cigarettes, which needs to be compared in future studies. There was some variation in cardiac FKBPL and CD31 expression within the groups from our in vivo study, which could be due to a small number of murine hearts per group that were processed for analyses. Increasing the number of mice per group, or performing Western blot on homeogenized tissue adjusted to a housekeeping protein may reduce variability.

## 4. Materials and Methods

### 4.1. Generation of EAC

E-Cigs utilise e-liquids that are heated to generate e-Cig vapour inhaled by users. To simulate a more physiological method of exposure, in preference of using e-liquid directly, we opted to heat the e-liquid as this will result in altered chemical composition to generate an aerosol [12]. For this study, EAC was generated using a KangerTech SUBOX mini e-cigarette device (KangerTech, Shenzhen, China) and tobacco flavoured e-liquid (Vape Empire, Sydney, NSW, Australia), both with (18 mg/mL), and without (0 mg/mL) nicotine. As a vehicle control, EAC was also generated from a stock solution composed of 80% propylene glycol and 20% vegetable glycerine (PG/VG) without tobacco flavour—the base composition of the e-liquid used for this study. The e-cigarette device was set at 30 W, and the air pump was simultaneously switched on for 5 s bursts, with 20 s to rest in between bursts. This setup created a vacuum trap that drew e-cigarette smoke into a 25 cm^2^ flask where the vaporised condensate was collected (Figure 7). The freshly generated condensate was rested upon dry ice for a minimum of 30 min before diluting to the final working concentrations and used immediately.

### 4.2. Cell Culture and Treatment Models 

HCAECs (Cell Applications, San Diego, CA, USA) were cultured in Endothelial Cell Growth Medium (Cell Applications, San Diego, CA, USA) and used from passages 1–10 in this current study. To study the metabolic process of lung tissue, we used A549 cells to model alveolar Type II pulmonary epithelium. A549 cells, human alveolar basal epithelial cell line from adenocarcinoma (A549; ATCC, Manassas, Virginia, USA), were cultured in DMEM (Thermo Fisher Scientific, Gibco, Waltham, MA, USA) supplemented with 10% foetal bovine serum (FBS) at 37 °C in a humidified atmosphere containing 5% CO_2_. A549 cells were used from passages 3–11 in this study.

A monoculture and indirect co-culture model using undiluted conditioned media (reviewed in Vis et al., 2020 [65]) treatment were utilized for this study. The monoculture involved direct treatment of the HCAECs with the EAC for 24 h. For the co-culture model, the A549 cells were seeded at 1 × 10^5^ cells per well in a 12-well plate and exposed to the EAC for 24 h. HCAEC were then exposed to the conditioned media (100%) obtained from the EAC-exposed A549 cells for an additional 24 h.

### 4.3. Cytotoxicity Assay

HCAEC were seeded at a concentration of 1 × 10^4^ cells per well in a 96-well plate and treated with EAC or conditioned media for 24 h. HCAEC not exposed to EAC or EAC-containing A549 conditioned media was used as the negative control. Following treatment, MTT reagent (10 µL of 5 mg/mL MTT; Sigma Aldrich, Castle Hill, NSW, Australia) was added to the media and cells were incubated for 3 h. Following incubation, the MTT/media mix was removed, cells were then washed with PBS before the addition of dimethyl sulfoxide (DMSO; 100 µL) to each well and absorbance at 565 nm was measured. Results were expressed as a percentage of negative control indicative of cell viability. 

### 4.4. Intracellular Reactive Oxygen Species (ROS) Assay 

HCAEC were seeded in a 96-well plate and treated with EAC or conditioned media for 24 h as described above. HCAEC not treated with EAC was used as the negative control, and cells treated with hydrogen peroxide (H_2_O_2_) were used as a positive control. Following treatment, the cells were incubated with 2′,7′-dichlorodihydrofluorescein diacetate (H_2_DCFDA) stain, and ROS level was determined as previously described [66]. Results were expressed as a percentage of negative control ROS activity. 

### 4.5. Enzyme-Linked Immunosorbent Assay (ELISA)

HCAEC were seeded in a 96-well plate and treated for 24 h with EAC. HCAEC not treated with EAC was used as the negative control. After treatment, ELISA was performed on the cells as previously described to determine the expression of the markers, VCAM-1 and ICAM-1 [66]. Cotinine concentration was measured in plasma using an ELISA kit (Abnova, Taipei, Taiwan) as per the manufacturer’s instructions.

### 4.6. Animal Exposure

Seven-week-old Balb/c female mice (n = 28) purchased from Animal Resource Centre (Perth, Western Australia, Australia) were housed in a 12 h light:12 h dark cycle with food and water available ad libitum. Following one week of acclimatisation, the mice in the same home cages were randomly assigned into three treatment groups (n = 9–10 per group) and exposed to ambient air (Sham): e-Cig aerosol generated from tobacco flavoured e-liquid with nicotine (18 mg/mL) or without nicotine (0 mg/mL). Each group was subjected to their respective treatment in a 9 L chamber filled with e-Cig aerosol in two fifteen-minute intervals with a five-minute aerosol free period in between, twice daily. Treatment conditions were based on previous maternal studies equating this exposure period to the smoke from two tobacco cigarettes. Tissue analysis on a subset of samples was performed in a double-blind manner, with group code only revealed during data analysis [67]. After 12 weeks of exposure, the mice were sacrificed, the left ventricle carefully excised, and snap-frozen in liquid nitrogen. 

The human relevance of exposure to nicotine-containing e-Cig aerosol in this model has been characterised by the serum cotinine levels, a stable nicotine metabolite, measured 16 to 20 h post last exposure to sham/E-Cig aerosol [40]. Serum cotinine level were as follows sham: 3.31 ± 0.386 ng/mL; 18 mg:17.41 ± 5.138 ng/mL; 0 mg: 5.97 ± 2.94 ng/mL. Additionally, previous studies have reported similar nicotine delivery volumes between e-Cigs and tobacco cigarettes (mean 1.3 mg e-Cig; 0.5–1.5 mg tobacco cigarette) [68]. These comparisons justify the comparison of cotinine levels between e-Cigs and cigarette users, in addition to the human relevance of mouse model illuminating the effects of e-Cig use. All animal experimental procedures were conducted in accordance with the guidelines described by the Australian National Health and Medical Research council code of conduct for animals with approval from the University of Technology Sydney Animal Care and Ethics Committee (ETH15-0025). 

### 4.7. Immunohistochemistry of the Heart Tissue 

The frozen left ventricles (LV) were halved, embedded in OCT, and sectioned (10 µm) using a Cryostat NX70 (Thermo Fisher Scientific, Gibco, Waltham, MA, USA). Slides were adhered onto gelatin-coated slides by air drying for 20 min before they were fixed in 10% formalin at −20 °C in the freezer for 20 min. Slides were washed in PBST (phosphate buffer saline + 0.1 Tween-20), incubated in blocking buffer (3% Goat serum diluted in 1% BSA in PBST–PBS with 0.1% Triton-X) for 1 h at room temperature before incubation with rabbit anti-FKBPL polyclonal antibody (1:100, Proteintech, Manchester, UK) and mouse anti-CD31 monoclonal antibody (1:100, Proteintech, Manchester, UK) in a humidity chamber. The sections were then washed with PBST (3 times over 15 min), incubated with donkey anti-rabbit AlexaFlour 488 and goat anti-mouse Alexfluor 594 (Abcam, Cambridge, UK) at 1:500 dilution, and counterstained with DAPI (Thermo Fisher Scientific, Gibco, Waltham, MA, USA; 1:20,000) at room temperature for 1 hr. Three images per section were captured at 20× magnification using an Olympus BX51 fluorescence microscope with an Olympus DP73 camera at varying exposure times (DAPI: 50 ms; FKBPL: 100 ms; CD31: 100 ms). ImageJ 1.53a was used to calculate the mean greyscale value of the fluorescent intensity of FKBPL and CD31 where values were normalised to the SHAM group as previously described [69,70]. To assess the validity of the immunohistochemistry staining, a negative control containing no primary antibody was used for each staining group.

### 4.8. Reverse Transcription-Polymerase Chain Reaction (RT-qPCR)

Total RNA was extracted from the other half of the LV by homogenisation in TRISURE (Bioline, Australia) using 1.4 mm zirconium oxide beads (Precellys, Bertin Technologies, Montigny-le-Bretonneux, France). Total RNA was then reverse transcribed using a Tetro cDNA synthesis kit (Bioline, Eveleigh, NSW, Australia) before qPCR was performed using SensiFAST SYBR No-ROX Kit (Bioline, Eveleigh, NSW, Australia) using the primers listed in Table 1. Total mRNA expression levels were calculated using the 2^−ΔΔCT^ method using β-actin as the reference gene [69].

### 4.9. Tethered Bilayer Lipid Membrane (tBLMs) Assay 

Gold-coated microscope slides with a monolayer coating of 10% benzyl disulphide eleven-oxygen-ethylene-glycol reservoir linkers with a C20 phytanyl group as ‘tethers’ and 90% four-oxygen-ethylene-glycol reservoir linkers with a terminal OH group as ‘spacers’ were purchased from SDx Tethered Membranes Pty Ltd., Sydney, Australia. A lipid bilayer was then anchored to the slides using a solvent-exchange technique that employed 3 mM ethanolic solutions of 1,2-Dioleoyl-sn-glycero-3-phosphocholine (DOPC) (Avanti Polar Lipids Inc., Alabaster, AL, USA) [70]. The solvent used for the exchange was 100 mM NaCl 10 mM Tris buffer at pH 7. Dilutions of the EAC used this same buffer. Measurements of membrane conductance were done using swept frequency electrical impedance spectroscopy using an applied potential of 25 mV peak-to-peak, ranging from 0.1 Hz to 2000 Hz, delivered using a Tethapod™ electrical impedance spectrometer (SDx Tethered Membranes Pty Ltd., Sydney, Australia). The data from the impedance and phase profiles were fitted to an equivalent circuit consisting of a constant phase element, representing the imperfect capacitance of the tethering gold electrode and reservoir region, in series with a resistor/capacitor representing the lipid bilayer and a resistor, to represent the impedance of the surrounding electrolyte solution, as described previously [71]. A proprietary adaptation of a Levenberg–Marquardt fitting routine incorporated into the TethaQuick™ software v2.0.56 (SDx Tethered Membranes Pty Ltd., Sydney, Australia) was used to fit the data.

### 4.10. Statistical Analysis 

All results are expressed as a mean ± SEM. The data was checked for normal distribution before parametric (one-way ANOVA) or non-parametric tests (Kruskal-Wallis) with post-hoc multiple comparison tests were used. GraphPad Prism v8.00 (IBM, Boston, MA, USA) was used to analyse the results. Results with *p* < 0.05 were considered significant.

## 5. Conclusions

Whilst the long-term adverse effects of e-Cig use on cardiovascular health are yet unknown, this study demonstrated that e-Cig condensates are associated with an increase in endothelial cell oxidative stress, inflammation, and cytotoxicity. This can impair endothelial cell integrity, lead to the restricted angiogenesis in the heart, and result in atherosclerosis and subsequently CVD. 

## Figures and Tables

**Figure 1 ijms-24-06378-f001:**
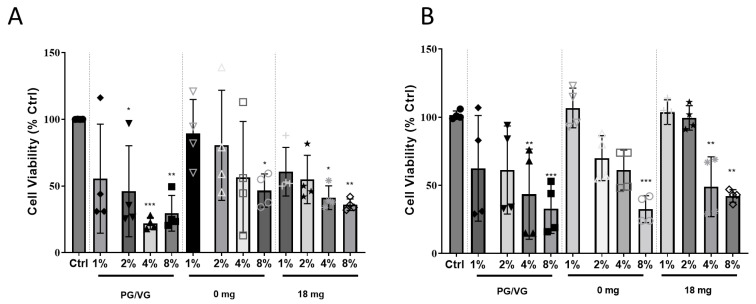
Cell viability in HCAEC exposed to (**A**) Direct effects of EAC. MTT Assay was performed on HCAEC after exposure to various concentration of EAC generated from: (i) a PG/VG solution (non-flavoured), (ii) 0 mg nicotine (tobacco flavoured), and (iii) 18 mg nicotine (tobacco flavoured) for 24 h. (**B**) Indirect effects of EAC. A549 epithelial lung cells were exposed to EAC under the same conditions. Media from the treated A549 cells were then used to treat HCAEC on a separate plate for 24 h before cell viability was assessed via MTT assay. Results are expressed as mean ± SEM (n = 4 biological replicates). One-way ANOVA with Bonferroni post-tests was used for statistical analysis; * *p* < 0.05, ** *p* < 0.01, *** *p* < 0.001 versus Ctrl.

**Figure 2 ijms-24-06378-f002:**
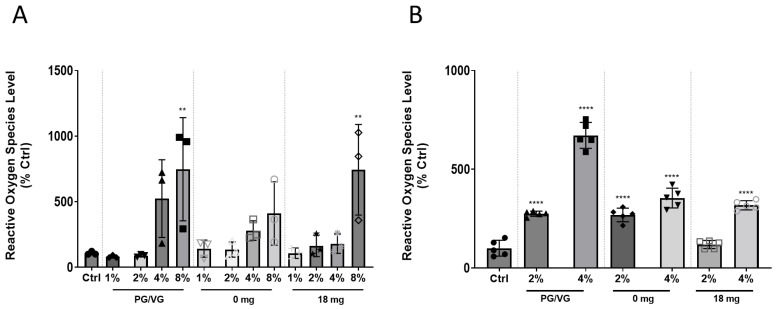
Reactive oxygen species levels in HCAEC after (**A**) Direct EAC exposure. ROS levels were measured in HCAEC after exposure to various concentration of EAC generated from: (i) a PG/VG standard (non-flavoured), (ii) 0 mg nicotine (tobacco flavoured), and (iii) 18 mg nicotine (tobacco flavoured) at for 24 h Data shown is expressed as a mean ± SEM (n = 3 biological replicates). (**B**) Indirect effects of EAC. A549 epithelial lung cells were exposed to EAC under the same conditions. Media from the treated A549 cells were then used to treat HCAEC on a separate plate for 24 h before a DCF assay was performed. Data shown is expressed as a mean ± SEM (n = 5 biological replicates). One-way ANOVA with Bonferroni post-tests was used for statistical analysis, ** *p* < 0.01; **** *p* < 0.0001 versus Ctrl.

**Figure 3 ijms-24-06378-f003:**
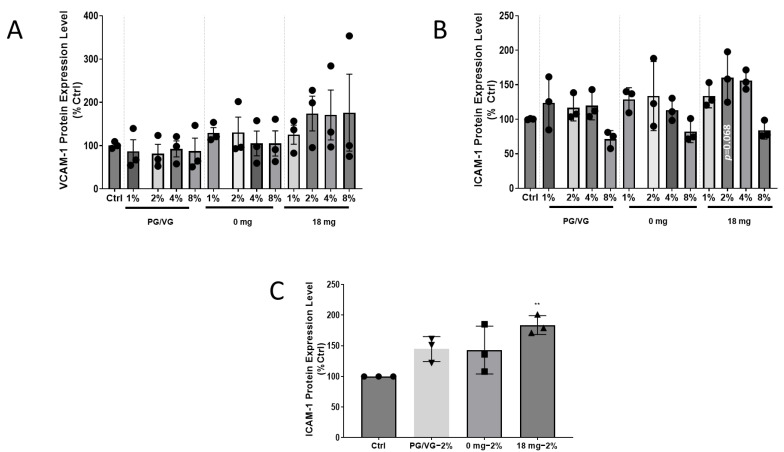
Expression of cellular adhesion molecules after exposure to EAC treatment. HCAEC were exposed to various concentrations of EAC generated from: (i) 0 mg nicotine (tobacco flavoured) and (ii) 18 mg nicotine (tobacco flavoured) for 24 h. (**A**) VCAM-1 protein expression. (**B**) ICAM-1 protein expression. (**C**) Indirect effects of EAC on ICAM-1 protein exposure. A549 epithelial lung cells were exposed to EAC under the same conditions. Media from the treated A549 cells were then used to treat HCAEC on a separate plate for 24 h before measuring ICAM-1 protein levels. Results are expressed as mean ± SEM (n = 3 biological replicates). One-way ANOVA with Bonferroni post-tests was used for statistical analysis, ** *p* < 0.01 versus Ctrl.

**Figure 4 ijms-24-06378-f004:**
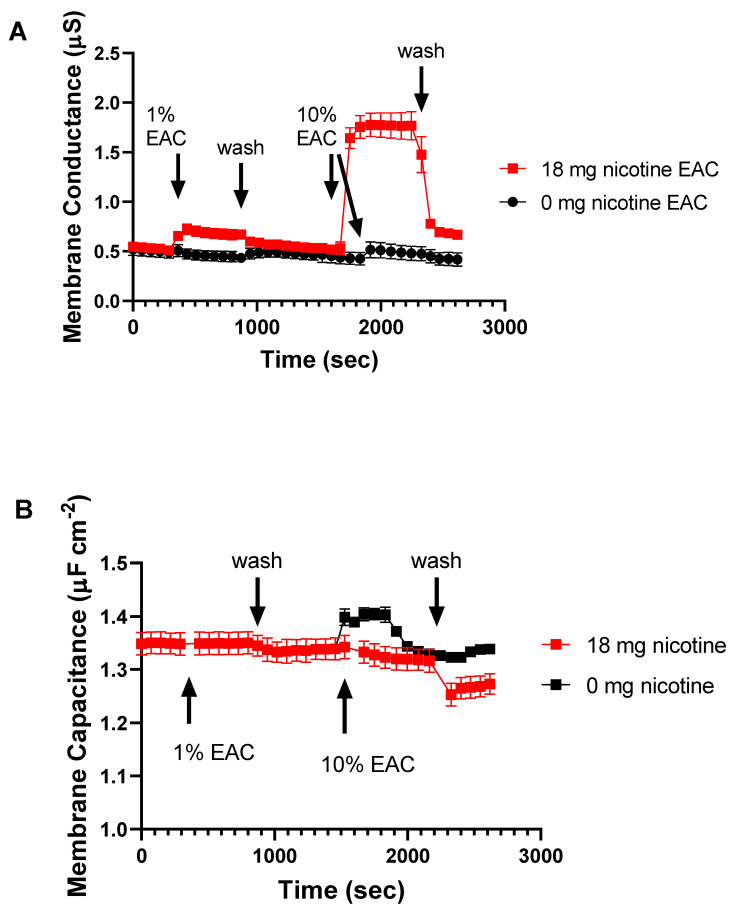
(**A**) Changes in membrane conduction of tethered bilayer lipid membranes (tBLM) in response to EAC (1% and 10%) in 100 mM NaCl 10 mM tris pH 7 buffer (n = 3). EAC solutions containing nicotine increase membrane conduction (membrane permeability). The effect of the nicotine-containing EAC rapidly falls away following a buffer wash. (**B**) In contrast, only minor changes of the membrane capacitances are observed in the same tBLMs, suggesting permeability changes aren’t related to large membrane structural changes.

**Figure 5 ijms-24-06378-f005:**
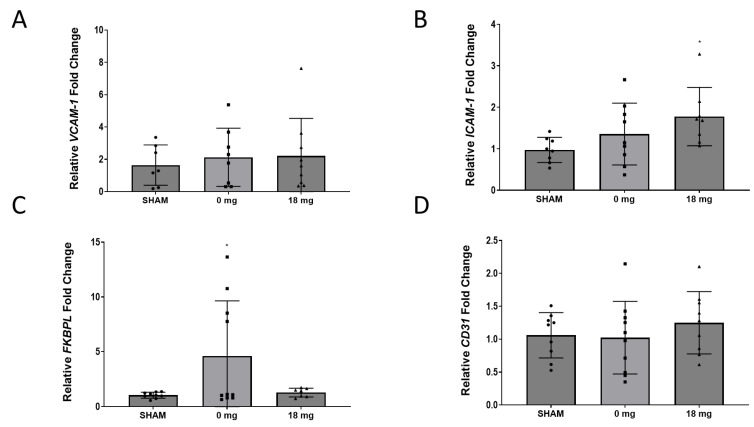
Cardiac VCAM1, ICAM1, and CD31 mRNA expression following treatment of mice with e-cigarettes with or without nicotine. RT-qPCR was performed on the left ventricle of mice exposed to ambient air (SHAM) or e-Cig aerosol (0 mg, 18 mg nicotine). (**A**) FKBPL. (**B**) CD31. (**C**) VCAM-1. (**D**) ICAM-1. All data expressed as mean fold change ± SEM (n = 5–9). One-way ANOVA with Bonferroni post-test was used for statistical analysis, * *p* < 0.05 versus Sham.

**Figure 6 ijms-24-06378-f006:**
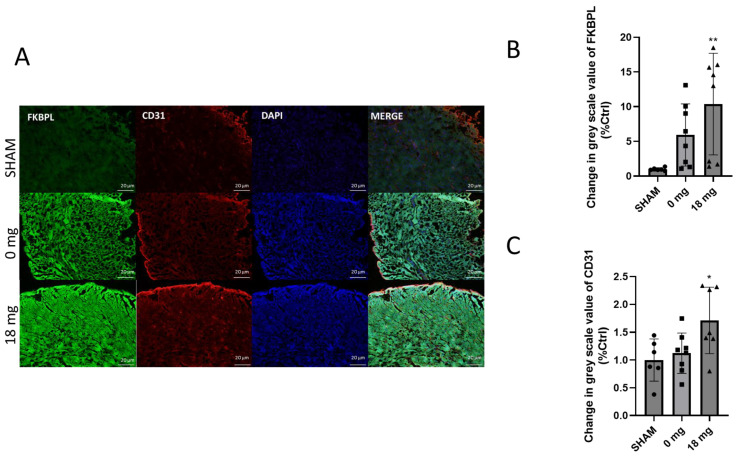
(**A**) Immunohistochemical on seven-week-old Balb/c female mice left ventricle sections (Scale bar = 20 μm). Mice were treated in 3 groups: SHAM (ambient air), 0 mg (no nicotine), and 18 mg (nicotine) treatment groups. Sections were stained for FKBPL (green), CD31 (red), and DAPI (blue) and images were taken at 20×. (**B**) FKBPL staining intensity was quantified as the mean greyscale value in three images per sample, ** *p* < 0.005 (SHAM vs. 18 mg). (**C**) CD31 staining intensity was quantified as the mean greyscale value in three images per sample, * *p* < 0.05 (SHAM vs. 18 mg). Results are expressed as mean ± SEM (n = 5–9) compared to SHAM. One-way ANOVA with Kruksal-Wallis post-tests was used for statistical analysis.

**Figure 7 ijms-24-06378-f007:**
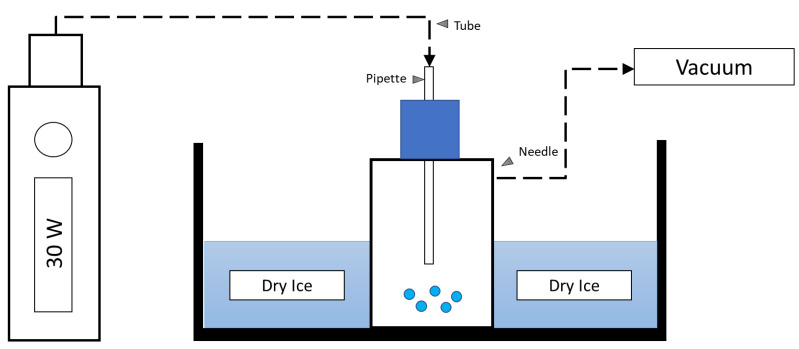
Experimental setup for e-cigarette aerosol condensate collection.

**Table 1 ijms-24-06378-t001:** qPCR primers and nucleotide sequence.

Primer Name	Primer Sequence (5′-3′)
*β-actin* (sense)	GATGTATGAAGGCTTTGGTC
*β-actin* (anti-sense)	TGTGCACTTTTATTGGTCTC
*ICAM-1* (sense)	CAGTCTACAACTTTTCAGCTC
*ICAM-1* (anti-sense)	CACACTTCACAGTTACTTGG
*VCAM-1* (sense)	ACTGATTATCCAAGTCTCTCC
*VCAM-1* (anti-sense)	CCATCCACAGACTTTAATACC
*CD31* (sense)	CATCGCCACCTTAATAGTTG
*CD31* (anti-sense)	CCAGAAACATCATCATAACCG
*FKBPL* (sense)	TCTCTCAGGGATCAGGAG
*FKBPL* (anti-sense)	TATTTAAGATTTGCTGGGCG

## Data Availability

All relevant data are contained within the article.

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
