# Peer review of "E-Cigarette Aerosol Condensate Leads to Impaired Coronary Endothelial Cell Health and Restricted Angiogenesis"

_ijms, 2023, doi:10.3390/ijms24076378_

Round 1

Reviewer 1 Report

The authors evaluate the effects of e-Cig aerosols (in the form of E-cigarette Aerosol Condensate) on endothelial cells. Both the direct effect on endothelial cells and the indirect effect in a coculture-like model system with alveolar cells, as well as several in vivo effect are evaluated.

The article is certainly interesting, however some major and minor revisions are required.

Introduction:

Well written and with an appropriate introductory reference to current knowledge. The reading is easy and the topics are well related to each other in a logical sequence.

Material and methods:

Minor revisions:

1)      Section 2.2

a.       A549 is an adenocarcinoma cell line, this should be clearly indicated as the wording "human alveolar epithelial cell line" can be misleading on this, in my opinion, limiting aspect.

b.       What was the density (cell/well), the culture time and the number of passages of the A549 from which the conditioned media with which the endothelial cells were treated was taken? This info must be added to ensure reproducibility of the experiment

2)      Section 2.3

a.       The authors state that the concentration of MTT used is 5 mg/ml. Is this the working concentration or is it the concentration of the stock solution which was then diluted in the culture medium? What was the working concentration of MTT added to the cells?

b.       Has the absorbance of the blank (well without cells) been subtracted?

c.       Was the MTT added to the culture medium in the presence of the treatment (condensed e.cig) or was this removed before exposure to the MTT? Has the direct interference of E-cigarette Aerosol Condensate on the MTT been tested? Is the absorbance of the MTT in the presence of only the E-cigarette Aerosol Condensate at the concentrations used comparable to that of the blank?

3)      Section 2.10

The authors state that "All results are expressed as a mean ± SEM", however in most of the figures they use a boxes and whiskers plot, which define minimum value, lower quartile, median, upper quartile, maximum value and not the value mean and the error on the mean value.

Results:

4)      Section 3.1

Is the observed reduction in cell viability due to a reduction in cell number or a reduction in cellular metabolic activity? Have they observed and counted the cells in order to gain information on these two different possibilities?

Have the viability between the two controls (HCAEC in appropriate culture medium versus the same cells exposed to DMEM in which A549 was cultured) been compared?

Is the viability the same between the two controls? Couldn't exposure to an unsuitable culture medium already depleted of essential components (for pre-culture with other cells) be the cause of further stress that exacerbates the effect of the condensed e-Cig?

Figure 2: "Results are expressed as mean ± SEM" should be correct. In both panels A) and B) it is stated that the number of experiments is 2 x 4 wells (meaning 8 trials), why is the number of data points shown on the graph different under different conditions?

5)      Section 3.2

Same as above: have the ROS production between the two controls (HCAEC in appropriate culture medium versus the same cells exposed to DMEM in which A549 was cultured) been compared?

Is the ROS production the same between the two controls? Couldn't exposure to an unsuitable culture medium already depleted of essential components (for pre-culture with other cells) be the cause of further stress that exacerbates the effect of the condensed e-Cig?

Figure 3: "Results are expressed as mean ± SEM" should be correct. Please check the number of data because there seems to be inconsistency between what is written in the legend and what is shown on the graph as in figure 2

6)      Section 3.3

Is the Icam protein level in the control significantly different from that observed in cells exposed to A549 conditioned medium (median approximately 100, vs. median approximately 60)?

Figure 4: "Results are expressed as mean ± SEM" should be correct.

Please check the number of data because there seems to be inconsistency between what is written in the legend and what is shown on the graph as in figure 2 as an example in panel C 12 data point in control, 6 in 0 mg-2%, 5 in 18 mg-2%

7)      Section 3.4

The authors show a change in membrane conductance, rapidly cleared after washing. Can the authors say anything regarding membrane capacitance? An analysis of the membrane capacity can provide further clarification on the interaction between membrane and nicotine: given that the effect on the conductance is transient and washed away by washing with a nicotine-free solution (not as happens for a channel which, inserting itself into the membrane , alters the permeability of the membrane), such a marked effect on the conductance could be explained by an alteration of the stability of the membrane with disorganization of the lateral packing of the lipid chains and variation of the membrane thickness, with consequent variation of the capacitance. The analysis of the variation of the membrane capacity is fundamental to define the effect of nicotine on the membrane.

8)      Section 3.6

The authors discuss a variation of expression based solely on the variation of gray levels of images acquired in immunofluorescence. Western blot experiments should be performed to confirm the quantitative data.

Figure 7 "Results are expressed as mean ± SEM" should be correct.

Please check the number of data because there seems to be inconsistency between what is written in the legend and what is shown on the graph

Discussion:

Well written, possible limitations of the study highlighted.

A further limitation of the study is that it remains largely descriptive of the effects of E-cig extract, which are examined descriptively without in-depth study of the molecular mechanisms leading to the observed effects.

Reviewer 2 Report

                This paper focuses on the the effects of e-cig condensate on endothelial cell behavior, more specifically presenting findings on inflammatory molecule expression, ROS, and viability. Membrane permeability data is also presented. The introduction covers appropriate literature and is sufficient context for the presented work. The methods are, in general, sufficiently detailed to understand and repeat the studies. The writing is clear, only minor editing is needed. The discussion is appropriately limited to the results as described, however, there are several concerns with the presentation and generation of the results, enumerated below:

Major

1) The negative control for the MTT is not clearly defined

2) The conditioned media experiment needs more detail (eg was it 100% CM or 50% or another mix). Additionally, conditioned media is distinct from co-culture, and this should be clear in the manuscript

3) More detail is required on the statistics, especially as the much of the data is groups of groups. For example, in Figure 4A, was the ANOVA run with 13 groups or with 3 sets of 5 groups?

3) The viability results presented in Figure 2 are concerning. As written, the results are two sets of quadruplicates, and it seems as if the two sets have wildly different results. This suggests either issues with the experimental setup or issues with the negative control (see above). Regardless, more experimental replicates are needed to assess the effects.

4) H2DCFDA staining is difficult to calibrate with varying cell numbers, what experiments were done to confirm cell loss was not a factor in these assays? Additionally, the data presented would benefit from presentation of the positive control.

5) How many experimental replicates are included in Figure 4A/B? N=2 is not sufficient for Figure 4C.

6) Figure 7 requires scalebars.

7) It is not clear if the images shown in Figure 7 are artifactual or showing differences in expression. What validation was done on the antibodies, were negative controls used, what were the imaging conditions (what camera type, exposure time, etc)? Are appropriate cells labeled in the sections (eg does CD31 localize to vessels, it looks like everything is illuminated). For the images presented, it isn’t clear that they are in focus (eg sham), although that is hard to tell based on image size. Improved methods and results presentation to address concerns about antibody specificity and the imaging conditions is critical to properly considering these results.

8) For the imaging, was 40X or 20X objective used?

Round 2

Reviewer 1 Report

The authors have certainly improved the article especially as regards the presentation of the data and the description of the materials and methods.

However, even though I know that MTT is a widely used assay for detecting viability, I would recommend the authors read articles like doi:10.3390/ijms222312827 which show that this is not always the correct assay to use, its limitations and that there may be interference from of substances present in the treatment, so it is certainly worth adding more rigorous controls (and also quick to do). According to the cited article (doi:10.3390/ijms222312827), the statement "To our knowledge, EAC is not expected to interfere with MTT reagent." it is not adequate.

I also know that the semi-quantitative densitometric analysis of an immunofluorescence image is performed, however it leaves me very perplexed that it is the only experimental data - affected by numerous tricky experimental and setup limitations - used to draw a conclusion which is an integral and important part of the article. A western blot (with an internal control such as an housekeeping protein) would be much more convincing. I am aware of the advantages offered by the immunofluorescence (that is not IHC) listed by the authors, but in this case, however, tha assay was only used to quantify protein expression in a semi-quantitative way.

Furthermore the authors have inserted a scale bar with reference to the number of pixels and not to micrometres. Given that the resolution of the image acquisition is not written, it remains difficult to trace the real scale used.

In conclusion, in my opinion the article could have been further improved. 

Author Response

Please see pdf attached for response.

Reviewer 2 Report

Critiques have been addressed.

Author Response

Thank you.

Round 3

Reviewer 1 Report

I think the authors have further improved the article.

Although they have sitll not done the western blot since experimental reasons make it impossible to perform it soon, having clearly expressed this limitation in the discussion, I think the article can now be accepted in the present form.